# DeepROCK: Error-controlled interaction detection in deep neural networks

## Abstract

The complexity of deep neural networks (DNNs) makes them powerful but also makes them challenging to interpret, hindering their applicability in error-intolerant domains. Existing methods attempt to reason about the internal mechanism of DNNs by identifying feature interactions that influence prediction outcomes. However, such methods typically lack a systematic strategy to prioritize interactions while controlling confidence levels, making them difficult to apply in practice for scientific discovery and hypothesis validation. In this paper, we introduce a method, called DeepROCK, to address this limitation by using knockoffs, which are dummy variables that are designed to mimic the dependence structure of a given set of features while being conditionally independent of the response. Together with a novel DNN architecture involving a pairwise-coupling layer, DeepROCK jointly controls the false discovery rate (FDR) and maximizes statistical power. In addition, we identify a challenge in correctly controlling FDR using off-the-shelf feature interaction importance measures. DeepROCK overcomes this challenge by proposing a calibration procedure applied to existing interaction importance measures to make the FDR under control at a target level. Finally, we validate the effectiveness of DeepROCK through extensive experiments on simulated and real datasets.

## 1 Introduction

Deep neural networks (DNNs) have emerged as a critical tool in many application domains, due in part to their ability to detect subtle relationships from complex data (Obermeyer & Emanuel, 2016). Though the complexity of DNNs is what makes them powerful, it also makes them challenging to interpret, leaving users with few clues about underlying mechanisms. Consequently, this "black box" nature of DNNs has hindered their applicability in error-intolerant domains such as healthcare and finance, because stakeholders need to understand why and how the models make predictions before making important decisions (Lipton, 2018).

To improve the interpretability of DNNs, many methods have been developed to reason about the internal mechanism of these models (Samek et al., 2021). These methods help to elucidate how individual features influence prediction outcomes by assigning an importance score to each feature so that higher scores indicate greater relevance to the prediction (Simonyan et al., 2013; Shrikumar et al., 2017; Lundberg & Lee, 2017; Sundararajan et al., 2017; Lu et al., 2021a). However, these univariate explanations neglect a primary advantage of DNNs, which is their ability to model complex interactions between features in a data-driven way. In fact, input features usually do not work individually within a DNN but cooperate with other features to make inferences jointly (Tsang et al., 2018a). For example, it is well established in biology that genes do not operate in isolation but work together in co-regulated pathways with additive, cooperative, or competitive interactions (Lu & Noble, 2021). Additionally, gene-gene, gene-disease, gene-drug, and gene-environment interactions are critical in explaining genetic mechanisms, diseases, and drug effects (Watson, 2022).

Several existing methods explain feature interactions in DNNs (Tsang et al., 2021). (See Section 3.2 for a detailed description of interaction detection methods.) Briefly, each such method detects interactions by inducing a ranking on candidate interactions from trained DNNs such that highly ranked interactions indicate greater detection confidence. Typically, this ranked list must be cut off at a certain confidence level for use in scientific discovery and hypothesis validation (Tsang et al., 2018a).

However, selecting this ranking threshold is typically under user control, subject to arbitrary choices and without scientific rigor. Worse still, existing methods are sensitive to perturbations, in the sense that even imperceivable, random perturbations of the input data may lead to dramatic changes in the importance ranking (Ghorbani et al., 2019; Kindermans et al., 2019; Lu et al., 2021b).

From a practitioner's perspective, a given set of detected interactions are only scientifically valuable if a systematic strategy exists to prioritize and select relevant interactions in a robust and error-controlled fashion, even in the presence of noise. Though many methods have been developed for interaction detection, we are not aware of previous attempts to carry out interaction detection while explicitly controlling the associated error rate. We propose to quantify the error via the false discovery rate (FDR) (Benjamini & Hochberg, 1995) and to use the estimated FDR to compare the performance of existing interaction detection methods. Informally, the FDR characterizes the expected proportion of falsely detected interactions among all detected interactions, where a false discovery is a feature interaction that is detected but is not truly relevant. (For a formal definition of FDR, see Section 2.2.) Commonly used procedures, such as the Benjamini–Hochberg procedure (Benjamini & Hochberg, 1995), achieve FDR control by working with p-values computed against some null hypothesis. In the interaction detection setting, for each feature interaction, one tests the significance of the statistical association between the specific interaction and the response, either jointly or marginally, and obtains a p-value under the null hypothesis that the interaction is irrelevant. These p-values are then used to rank the features for FDR control. However, FDR control in DNNs is challenging because, to our knowledge, the field lacks a method for producing meaningful p-values reflecting interaction importance in DNNs.

To bypass the use of p-values but still achieve FDR control, we draw inspiration from the model-X knockoffs framework (Barber & Candès, 2015; Candès et al., 2018). In this approach, the core idea is to generate "knockoff" features that perfectly mimic the empirical dependence structure among the original features but are conditionally independent of the response given the original features. These knockoff features can then be used as a control by comparing the feature importance between the original features and their knockoff counterparts to achieve error-controlled feature selection. In this paper, we apply the idea of a knockoff filter to DNNs and propose an error-controlled interaction detection method named DeepROCK (Deep inteRaction detectiOn using knoCKoffs). At a high level, DeepROCK makes two primary contributions. First, DeepROCK uses a novel, multilayer perceptron (MLP) architecture that includes a plugin pairwise-coupling layer (Lu et al., 2018) containing multiple filters, one per input feature, where each filter connects the original feature and its knockoff counterpart. As such, DeepROCK achieves FDR control via the knockoffs and maximizes statistical power by encouraging the competition of each feature against its knockoff counterpart through the pairwise-coupling layer. Second, we discover that naively using off-the-shelf feature interaction importance measures cannot correctly control FDR. To resolve this issue, DeepROCK proposes a calibration procedure applied to existing interaction importance measures to make the FDR under control at a target level. Finally, we have applied DeepROCK to both simulated and real datasets to demonstrate its empirical utility.

## 2 BACKGROUND

### 2.1 PROBLEM SETUP

Consider a supervised learning task where we have $n$ independent and identically distributed (i.i.d.) samples $\mathbf{X} = \{x_i\}_{i=1}^{n} \in \mathbb{R}^{n \times p}$ and $\mathbf{Y} = \{y_i\}_{i=1}^{n} \in \mathbb{R}^{n \times 1}$, denoting the data matrix with $p$-dimensional features and the corresponding response, respectively. The task is modeled by a black-box function $f : \mathbb{R}^p \mapsto \mathbb{R}$, parameterized by a deep neural network (DNN) that maps from the input $x \in \mathbb{R}^p$ to the response $y \in \mathbb{R}$. When modeling the task, the function $f$ learns non-additive feature interactions from the data, of which each interaction $\mathcal{I} \subset \{1, \cdots, p\}$ is a subset of interacting features. In this work, we focus on pairwise interactions, i.e., $|\mathcal{I}| = 2$. We say that $\mathcal{I}$ is a non-additive interaction of function $f$ if and only if $f$ cannot be decomposed into an addition of $|\mathcal{I}|$ subfunctions $f_i$, each of which excludes a corresponding interaction feature (Sorokina et al., 2008; Tsang et al., 2018a), i.e., $f(x) \neq \sum_{i \in \mathcal{I}} f_i(x_{\{1, \cdots, p\} \setminus i})$. For example, the multiplication between two features $x_i$ and $x_j$ is a non-additive interaction because it cannot be decomposed into a sum of univariate functions, i.e., $x_i x_j \neq f_i(x_j) + f_j(x_i)$. Assume that there exists a group of interactions $\mathcal{S} = \{\mathcal{I}_1, \mathcal{I}_2, \cdots\}$ such that conditional on interactions $\mathcal{S}$, the response $\mathbf{Y}$ is independent of interactions

in the complement $\mathcal{S}^c = \{1, \cdots, p\} \times \{1, \cdots, p\} \backslash \mathcal{S}$. Existing interaction detection methods (Tsang et al., 2021) induce a ranking on candidate interactions based upon the trained model $f$, where highly ranked interactions indicate more substantial detection confidence. Therefore, our goals are to (1) learn the dependence structure of $\mathbf{Y}$ on $\mathbf{X}$ so that effective prediction can be made with the fitted model and (2) achieve accurate interaction detection by identifying interactions in $\mathcal{S}$ with a controlled error rate.

## 2.2 FALSE DISCOVERY RATE CONTROL AND THE KNOCKOFF FILTER

DeepROCK measures the performance of an interaction detection method using the false discovery rate (FDR) (Benjamini & Hochberg, 1995). For a set of feature interactions $\widehat{S} \subset \{1, \cdots, p\} \times \{1, \cdots, p\}$ selected by some interaction detection method, the FDR is defined as

$$\text{FDR} = \mathbb{E}[\text{FDP}] \text{ with FDP} = \frac{|\widehat{S} \cap \mathcal{S}^c|}{|\widehat{S}|},$$

where $|\cdot|$ stands for the cardinality of a set. Though many methods have been proposed to achieve FDR control (Benjamini & Yekutieli, 2001; Storey et al., 2004; Abramovich et al., 2006; Fan et al., 2007; Wu, 2008; Clarke & Hall, 2009; Hall & Wang, 2010; Fan et al., 2012), most of these methods rely on p-values and hence cannot be directly adapted to the DNN setting.

In this paper, DeepROCK controls the FDR by leveraging the knockoffs framework (Barber & Candès, 2015; Candès et al., 2018), which was proposed in the setting of error-controlled feature selection. The core idea of this method is to generate knockoff features that perfectly mimic the empirical dependence structure among the original features. Briefly speaking, the knockoff filter achieves FDR control in two steps: (1) construction of knockoff features and (2) filtering using knockoff statistics. For the first step, the knockoff features are defined as follows:

**Definition 1** (Model-X knockoff (Candès et al., 2018)). The model-X knockoff features for the family of random features $\mathbf{X} = (X_1, \ldots, X_p)$ are a new family of random features $\tilde{\mathbf{X}} = (\tilde{X}_1, \ldots, \tilde{X}_p)$ that satisfy two properties:

1. $(\mathbf{X}, \tilde{\mathbf{X}})_{\text{swap}(\mathcal{S})} \stackrel{d}{=} (\mathbf{X}, \tilde{\mathbf{X}})$ for any subset $\mathcal{S} \subset \{1, \ldots, p\}$, where swap($\mathcal{S}$) means swapping $X_j$ and $\tilde{X}_j$ for each $j \in \mathcal{S}$ and $\stackrel{d}{=}$ denotes equal in distribution, and

2. $\tilde{\mathbf{X}} \perp\!\!\!\perp \mathbf{Y} | \mathbf{X}$, i.e., $\tilde{\mathbf{X}}$ is independent of response $\mathbf{Y}$ given feature $\mathbf{X}$.

According to Definition 1, the construction of the knockoffs must be independent of the response $\mathbf{Y}$. Thus, if we can construct a set $\tilde{X}$ of model-X knockoff features properly, then by comparing the original features with these control features, FDR can be controlled at target level $q$. In the Gaussian setting, *i.e.,* $\mathbf{X} \sim \mathcal{N}(0, \mathbf{\Sigma})$ with covariance matrix $\mathbf{\Sigma} \in \mathbb{R}^{p \times p}$, the model-X knockoff features can be constructed easily:

$$\tilde{\mathbf{X}}|\mathbf{X} \sim N\big(\mathbf{X} - \text{diag}\{\mathbf{s}\}\mathbf{\Sigma}^{-1}\mathbf{X}, 2\text{diag}\{\mathbf{s}\} - \text{diag}\{\mathbf{s}\}\mathbf{\Sigma}^{-1}\text{diag}\{\mathbf{s}\}\big) \qquad (1)$$

where $\text{diag}\{\mathbf{s}\}$ is a diagonal matrix with all components of $\mathbf{s}$ being positive such that the conditional covariance matrix in Equation 1 is positive definite. As a result, the original features and the model-X knockoff features constructed by Equation 1 have the following joint distribution:

$$(\mathbf{X}, \tilde{\mathbf{X}}) \sim \mathcal{N}\left(\begin{pmatrix} \mathbf{0} \\ \mathbf{0} \end{pmatrix}, \begin{pmatrix} \mathbf{\Sigma} & \mathbf{\Sigma} - \text{diag}\{\mathbf{s}\} \\ \mathbf{\Sigma} - \text{diag}\{\mathbf{s}\} & \mathbf{\Sigma} \end{pmatrix}\right). \qquad (2)$$

It is worth mentioning that the conventional knockoffs are restricted to the Gaussian settings, which may not be applicable in many practical settings. According, DeepROCK uses KnockoffGAN (Jordon et al., 2018), a commonly-used knockoff framework with no assumptions on the feature distribution. In principle, DeepROCK is generalizable to any existing non-Gaussian knockoff generation method, such as auto-encoding knockoffs (Liu & Zheng, 2018), deep knockoffs (Romano et al., 2020), or DDLK (Sudarshan et al., 2020).

With the constructed knockoff $\tilde{\mathbf{X}}$, feature importances are quantified by computing the knockoff statistics $W_j = g_j(Z_j, \tilde{Z}_j)$ for $1 \leq j \leq p$, where $Z_j$ and $\tilde{Z}_j$ represent feature importance measures

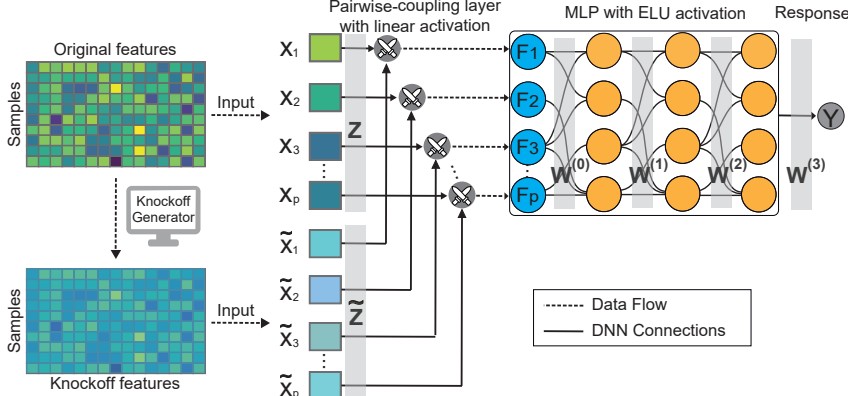

Figure 1: **Overview of DeepROCK.** DeepROCK is built upon an MLP with a plugin pairwise-coupling layer containing $p$ filters, one per input feature, where each filter connects the original feature and its knockoff counterpart. The filter weights $Z_j$ and $\tilde{Z}_j$ for the $j$-th feature and its knockoff counterpart are initialized equally for fair competition. The outputs of the filters are fed into a fully connected MLP with ELU activation.

for the $j$-th feature $X_j$ and its knockoff counterpart $\tilde{X}_j$, respectively, and $g_j(\cdot, \cdot)$ is an antisymmetric function satisfying $g_j(Z_j, \tilde{Z}_j) = -g_j(\tilde{Z}_j, Z_j)$. The knockoff statistics $W_j$ should satisfy a coin-flip property such that swapping an arbitrary pair $X_j$ and its knockoff counterpart $\tilde{X}_j$ only changes the sign of $W_j$ but keeps the signs of other $W_k$ ($k \neq j$) unchanged (Candès et al., 2018). A desirable property for knockoff statistics $W_j$'s is that important features are expected to have large positive values, whereas unimportant ones should have small symmetric values around 0.

Finally, the absolute values of the knockoff statistics $|W_j|$'s are sorted in decreasing order, and FDR-controlled features are selected whose $W_j$'s exceed some threshold $T$. In particular, the choice of threshold $T$ follows $T = \min \left\{ t \in \mathcal{W}, \frac{1+|\{j:W_j \leq -t\}|}{|\{j:W_j \geq t\}|} \leq q \right\}$ where $\mathcal{W} = \{|W_j| : 1 \leq j \leq p\} \setminus \{0\}$ is the set of unique nonzero values from $|W_j|$'s and $q \in (0, 1)$ is the desired FDR level specified by the user.

Note that the design of the knockoff filter depends on what types of discoveries are being subjected to FDR control. Specifically, conventional knockoff filters use feature-based knockoff statistics to control the feature-wise FDR, whereas DeepROCK designs interaction-based knockoff statistics and employs an interaction-specific selection procedure tailored for interaction-wise FDR control. (See Section 3 for more details.)

## 3 APPROACH

### 3.1 KNOCKOFF-TAILORED DNN ARCHITECTURE

DeepROCK integrates the idea of knockoffs with DNNs to achieve interaction detection with controlled FDR, as illustrated in Figure 1. Specifically, DeepROCK first generates the knockoffs $\tilde{\mathbf{X}} \in \mathbb{R}^{n \times p}$ from the input data $\mathbf{X} \in \mathbb{R}^{n \times p}$ by following the procedure described in Section 2.2. After concatenation, an augmented data matrix $(\mathbf{X}, \tilde{\mathbf{X}}) \in \mathbb{R}^{n \times 2p}$ is fed into the DNN through a plugin pairwise-coupling layer containing $p$ filters, $\mathbf{F} = (F_1, \cdots, F_p) \in \mathbb{R}^p$, where the $j$-th filter connects feature $X_j$ and its knockoff counterpart $\tilde{X}_j$. The filter weights $\mathbf{Z} \in \mathbb{R}^p$ and $\tilde{\mathbf{Z}} \in \mathbb{R}^p$ are initialized equally and compete against each other through pairwise connections during training. Thus, intuitively, $\mathbf{Z}_j$ being much larger than $\tilde{\mathbf{Z}}_j$ in magnitude provides some evidence that the $j$-th feature is important, possibly indicating the involvement of important interactions, whereas similar values of $\mathbf{Z}_j$ and $\tilde{\mathbf{Z}}_j$ indicate that the $j$-th feature is not important. In addition to the competition of each feature against its knockoff counterpart, we also encourage competition among features by using a linear activation function in the pairwise-coupling layer.

The outputs of the filters are then fed into a fully connected multilayer perceptron (MLP) with $L$ hidden layers to learn a mapping to the response $\mathbf{Y}$. In this work, we use an MLP with $L = 3$ hidden layers, as illustrated in Figure 1, where the choice of layer number is only for illustration purposes. We let $p_l$ be the number of neurons in the $l$-th layer of the MLP, where $p_0 = p$, and we let $\mathbf{W}^{(0)} \in \mathbb{R}^{p \times p_1}$, $\mathbf{W}^{(1)} \in \mathbb{R}^{p_1 \times p_2}$, $\mathbf{W}^{(2)} \in \mathbb{R}^{p_2 \times p_3}$, and $\mathbf{W}^{(3)} \in \mathbb{R}^{p_3 \times 1}$ be the weight matrices connecting successive layers in the model. In this way, the response $\mathbf{Y}$ can be represented as

$$
\begin{aligned}
\mathbf{h}^{(0)} &= \mathbf{F}, \\
\mathbf{h}^{(l)} &= \text{ELU} \left( \mathbf{W}^{(l-1)} \mathbf{h}^{(l-1)} + \mathbf{b}^{(l-1)} \right), \text{ for } l = 1, \cdots, L \\
\mathbf{Y} &= \mathbf{W}^{(L)} \mathbf{h}^{(L)} + \mathbf{b}^{(L)}
\end{aligned}
\tag{3}
$$

where $\text{ELU}(\cdot)$ is an exponential linear unit, and $\mathbf{b}^{(l)} \in \mathbb{R}^{p_l}$ denotes the bias vector in the $l$-th layer.

## 3.2 FEATURE INTERACTION IMPORTANCE

As a necessary step towards FDR estimation, DeepROCK aims to induce a ranking on candidate interactions such that highly ranked interactions indicate greater detection confidence. For notational simplicity, we index the feature, involving both original features and knockoffs, by $\{1, 2, \cdots, 2p\}$, where $\{1, \cdots, p\}$ and $\{p + 1, \cdots, 2p\}$ are the indices with correspondence for original features and their knockoff counterparts, respectively. We hereafter denote $\mathbf{S}^{\text{2D}} = \left[ s_{ij}^{\text{2D}} \right]_{i,j=1}^{2p} \in \mathbb{R}^{2p \times 2p}$ as the importance measure for feature interactions.

We consider two representative variants of importance measures to demonstrate DeepROCK's flexibility. We first use the underline{model-based} importance that explains the relationship between features and responses derived from the model weights (Tsang et al., 2018a;b; Yang et al., 2021; Cui et al., 2020; Liu et al., 2020; Badre & Pan, 2022), which further decompose into two factors: (1) the relative importance between the original feature and its knockoff counterpart, encoded by concatenated filter weights $\mathbf{Z}^{\text{Agg}} = (\mathbf{Z}, \tilde{\mathbf{Z}}) \in \mathbb{R}^{2p}$, and (2) the relative importance among all $p$ features, encoded by the weight matrix $\mathbf{W}^{(0)} \in \mathbb{R}^{p \times p_1}$ and the aggregated weights $\mathbf{W}^{\text{Agg}} = \mathbf{W}^{(1)} \mathbf{W}^{(2)} \mathbf{W}^{(3)} \in \mathbb{R}^{p_1}$. (See Garson (1991) for theoretical insights regarding $\mathbf{W}^{\text{Agg}}$.) Inspired by Tsang et al. (2018a), we define the model-based feature interaction importance as

$$
s_{ij}^{\text{2D}} = \left( \mathbf{Z}_i^{\text{Agg}} \mathbf{W}_i^{\text{INT}} \odot \mathbf{Z}_j^{\text{Agg}} \mathbf{W}_j^{\text{INT}} \right)^T \mathbf{W}^{\text{Agg}}
\tag{4}
$$

where $\mathbf{W}^{\text{INT}} = (\mathbf{W}^{(0)T}, \mathbf{W}^{(0)T})^T \in \mathbb{R}^{2p \times p_1}$ and $\mathbf{W}_j^{\text{INT}} \in \mathbb{R}^{p_1}$ denotes the $j$-th row of $\mathbf{W}^{\text{INT}}$.

Additionally, we use the underline{instance-based} importance that explains the relationships between features and responses across all samples (Cui et al., 2019; Lundberg et al., 2020; Tsang et al., 2020; Janizek et al., 2021; Sundararajan et al., 2020; Chang et al., 2022; Lerman et al., 2021; Zhang et al., 2021). Inspired by Janizek et al. (2021), we define the instance-based feature interaction importance as

$$
s_{ij}^{\text{2D}} = \sum_{x \in \mathbf{X}} \int_{x'} (x_i - x_i')(x_j - x_j') \times \int_{\beta=0}^{1} \int_{\alpha=0}^{1} \nabla_{i,j} \mathbf{Y}(x' + \alpha\beta(x - x')) d\alpha d\beta dx'
\tag{5}
$$

where $\nabla_{i,j} \mathbf{Y}(x)$ calculates the second-order Hessian of the response $\mathbf{Y}$ with respect to the input $x$.

We initially experimented with FDR control on the induced ranking by naively using model-based or instance-based feature interaction importance measures. However, we discovered that naively using these existing importance measures does not correctly control the FDR; see Figure 2 and the discussion in Section 4.1. We hypothesized that the problem lies in violating the knockoff's assumption that knockoff feature interaction scores have the same distribution as the irrelevant feature interaction scores. Intuitively, the interaction between two marginally important features naturally has a higher importance score than random interactions, even though they are all false. To resolve this issue, DeepROCK proposes a calibration procedure to apply on top of existing interaction importance measures. The calibrated interaction between the $i$-th and $j$-th features is defined as

$$
\mathbf{S}_{ij} = \frac{\left| S_{ij}^{\text{2D}} \right|}{\sqrt{\left| S_i^{\text{1D}} \cdot S_j^{\text{1D}} \right|}}
\tag{6}
$$

where $\mathbf{S}^{1D} = \left[ s_j^{1D} \right]_{j=1}^{2p} \in \mathbb{R}^{2p}$ denotes the univariate feature importance measure compatible with the corresponding interaction measure. Specifically, by following Lu et al. (2018), we define the model-based univariate feature importance as

$$\mathbf{S}^{1D} = (\mathbf{Z} \odot \mathbf{W}^{1D}, \tilde{\mathbf{Z}} \odot \mathbf{W}^{1D}) \tag{7}$$

where $\mathbf{W}^{1D} = \mathbf{W}^{(0)} \mathbf{W}^{\text{Agg}} \in \mathbb{R}^p$ and $\odot$ denotes entry-wise matrix multiplication. Additionally, by following Erion et al. (2021), we define the instance-based univariate feature importance as

$$s_j^{1D} = \sum_{x \in \mathbf{X}} \int_{x'} (x_j - x'_j) \times \int_{\alpha=0}^1 \nabla_j \mathbf{Y}(x' + \alpha(x - x')) d\alpha dx' \tag{8}$$

where $\nabla_j \mathbf{Y}(x)$ calculates the first-order gradient of the response $\mathbf{Y}$ with respect to the input $x$.

### 3.3 FDR CONTROL FOR INTERACTIONS

After calculating the feature interaction importance using Equation 6, we denote the resultant set of interaction importance scores as $\Gamma = \{S_{ij} | i < j, i \neq j - p\}$. We sort $\Gamma$ in decreasing order and select interactions whose importance $\Gamma_j$ exceed some threshold $T$ such that the selected interactions are subject to a desired FDR level $q \in (0,1)$. A complication arises from the heterogeneous interactions, containing original-original interactions, original-knockoff interactions, knockoff-original interactions, and knockoff-knockoff interactions. Following Walzthoeni et al. (2012), the choice of threshold $T$ follows:

$$T = \min \left\{ t \in \mathcal{T}, \frac{|\{j : \Gamma_j \geq t, j \in \mathcal{D}\}| - 2 \cdot |\{j : \Gamma_j \geq t, j \in \mathcal{DD}\}|}{|\{j : \Gamma_j \geq t\}|} \leq q \right\} \tag{9}$$

where $\mathcal{D}$ and $\mathcal{DD}$ refer to the set of interactions, each of which contain at least one knockoff feature and both knockoff features, respectively. And $\mathcal{T}$ is the set of unique nonzero values in $\Gamma$.

## 4 RESULTS

### 4.1 PERFORMANCE ON SIMULATED DATA

We begin by using synthetic data to evaluate the performance of DeepROCK from the following two perspectives: (1) Can DeepROCK accurately estimate the FDR among detected interactions? (2) How effective is DeepROCK in detecting true interactions with a controlled FDR?

#### 4.1.1 EXPERIMENTAL SETUP

For these experiments, we used a test suite of 10 simulated datasets (Table A.1) that contain a mixture of univariate functions and multivariate interactions with varying order, strength, and nonlinearity. Because we aim to detect pairwise interactions, we decompose high-order interaction functions (*e.g.,* $F(x_1, x_2, x_3) = x_1 x_2 x_3$) into pairwise interactions (*e.g.,* $(x_1, x_2)$, $(x_1, x_3)$, and $(x_2, x_3)$) as the ground truth. Following the settings reported in Tsang et al. (2018a), we used a a sample size of $n = 20,000$, evenly split into training and test sets. Additionally, we set the number of features to $p = 30$, where all features were sampled from a continuous uniform distribution $U(0,1)$. As shown in Table A.1, only 10 out of 30 features contribute to the corresponding response, and the remaining serve as noise to complicate the task. For each simulated dataset, we repeated the experiment 20 times, where each repetition involves data generation and neural network training with different random seeds. For all simulation settings, we set the target FDR level to $q = 0.2$.

#### 4.1.2 SIMULATION RESULTS

We first evaluated the impact of the calibration procedure on existing feature interaction importance measures, each inducing a ranking on candidate interactions in terms of their importance. The ranking performance is measured by the area under the receiver operating characteristic curve (AUROC) with respect to the gold standard list of interactions. As shown in Figure 2(A), calibrated feature interaction importance measures achieve comparable performance to the ones without calibration

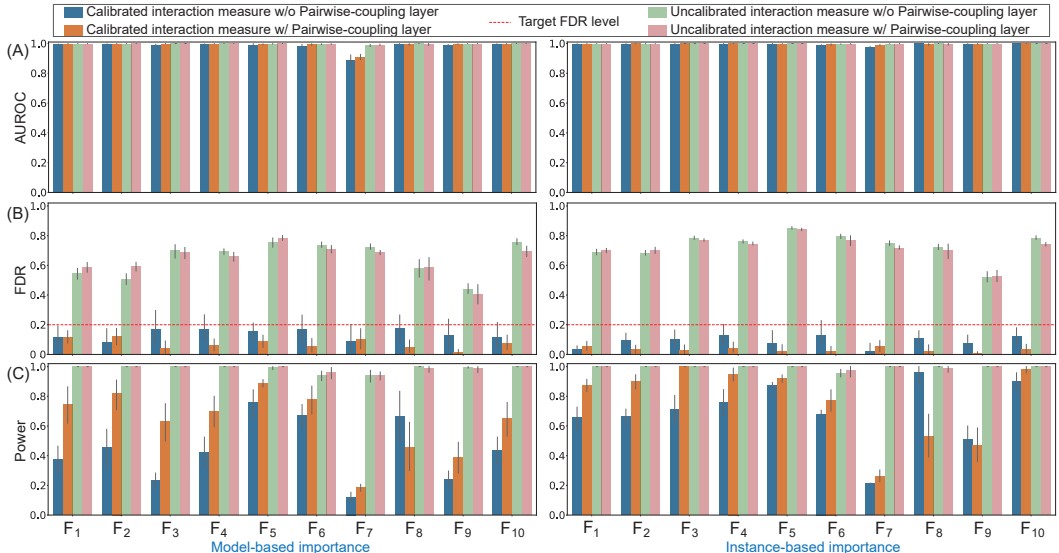

Figure 2: Evaluating DeepROCK on a test suite of 10 simulated datasets in terms of AUROC, FDR, and power. Error bars correspond to the 95% confidence interval of AUROC, FDP, and power across 20 repetitions. (A) The proposed calibration procedure achieves comparable performance in ranking important interactions, as measured by AUROC. (B) Existing feature interaction importance measures cannot correctly control FDR, whereas the proposed calibration procedure can control FDR below the target FDR level. (C) Adopting the pairwise-coupling layer achieves FDR control with much higher power.

in ranking important interactions, as measured by AUROC. comparable to what? The only exception is a very challenging function $F_7$ using the model-based importance, but the instance-based importance still works well in $F_7$ with calibration.

Given the comparable AUROC, we investigated whether the top-ranked interactions could achieve controlled FDR. We discover, surprisingly, that naively using existing feature interaction importance measures without calibration does not correctly control the FDR. In comparison, existing feature interaction importance measures with calibration consistently controls the FDR much below the target FDR level. As shown in Figure 2(B), the results suggest that DeepROCK tends to be conservative, and DeepROCK could potentially gain statistical power simply by improving FDR estimation.

Finally, we examined the necessity of adopting the plugin pairwise-coupling layer. As shown in Figure 2(C), after adopting the pairwise-coupling layer, DeepROCK consistently achieves FDR control with much higher power. The results confirm that the pairwise-coupling layer directly encourages competition between original and knockoff features (Lu et al., 2018). It is worth mentioning that the feature interaction importance without calibration achieves nearly perfect power at the expense of uncontrolled FDR. However, without a controlled FDR, the seemingly perfect power becomes suspicious and meaningless.

## 4.2 REAL DATA ANALYSIS

In addition to the simulated datasets presented in Section 4.1, we also demonstrate the practical utility of DeepROCK on two real applications. For both studies the target FDR level is set to $q = 0.1$.

### 4.2.1 APPLICATION TO DROSOPHILA ENHANCER DATA

We first applied DeepROCK to investigate the relationship between enhancer activity and DNA occupancy for transcription factor (TF) binding and histone modifications in *Drosophila* embryos. We used a quantitative study of DNA occupancy for $p_1 = 23$ TFs and $p_2 = 13$ histone modifications with labelled enhancer status for $n = 7,809$ genomic sequence samples in blastoderm *Drosophila* embryos (Basu et al., 2018). The enhancer status for each genomic sequence is binarized as the

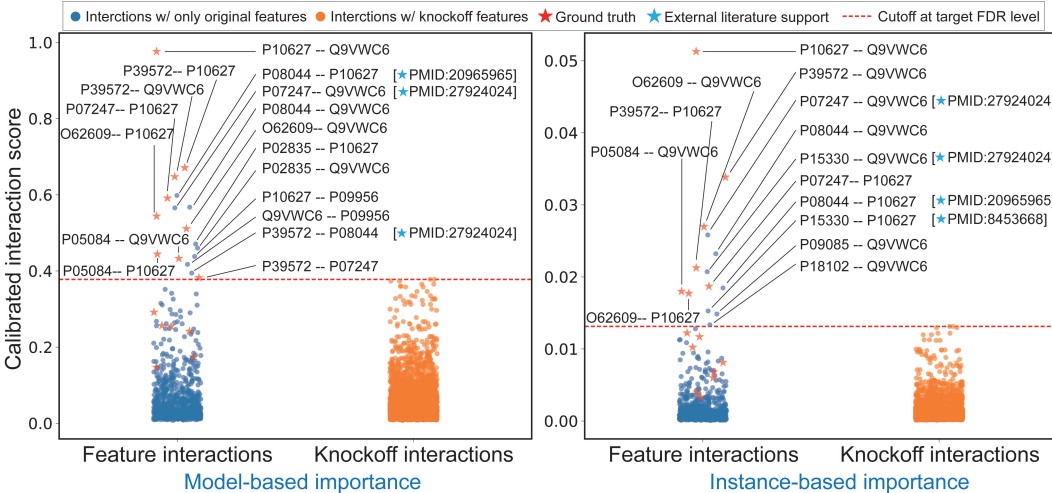

Figure 3: Applying DeepROCK to the Drosophila enhancer data using both model-based and instance-based importance measures. Though DeepROCK only induces a ranking on feature interactions, knockoff-involving interactions are also provided for comparison. The feature interactions identified by DeepROCK at the target FDR level are labeled with the corresponding TFs in terms of their UniProt identifiers. The red stars indicate the well-characterized interactions in early *Drosophila* embryos as ground truth. The blue stars indicate the interactions reported by the database containing the experimentally verified interactions and are supported by literature evidence as indicated by the accompanying PubMed identifiers.

response, depending on whether the sequence drives patterned expression in blastoderm embryos. As features to predict enhancer status, the maximum value of normalized fold-enrichment (Li et al., 2008) is used for each TF or histone modification.

We first evaluated the identified TF-TF interactions by DeepROCK at the target FDR level using both model-based and instance-based importance measures. The evaluations are from three perspectives. First, we compared the identified interactions against a list of the well-characterized interactions in early *Drosophila* embryos summarized by Basu et al. (2018) as ground truth. If DeepROCK does a good job identifying important interactions subject to FDR control, then the identifications should overlap heavily with the ground truth list. As shown in Figure 3, in the two different settings, 9 out of 17 and 7 out of 14 interactions identified by DeepROCK overlap with ground truth list, respectively. Second, we investigated the identified interactions that were not included in the ground truth list. In the two different settings, 3 out of 8 and 4 out of 7 remaining interactions, respectively, are reported by a database containing the experimentally verified interactions (Liska et al., 2022). These experimentally verified interactions are also supported by literature evidence, whose PubMed identifiers are shown in Figure 3. Finally, we scrutinized the remaining identified interactions without ground truth or literature evidence support, and we found that transitive effects can explain these interactions. Intuitively, if there is a strong interaction between TF1 and TF2, and between TF2 and TF3, then a high interaction score will also be expected between TF1 and TF3, even if there is no direct interaction between them. For example, the interaction between the TF's Snail (UniProt ID: P08044) and Zelda (UniProt ID: Q9VWC6), which is identified in both settings, can be regarded as a transitive interaction between two well-supported interactions: (1) the interaction between Snail and Twist (UniProt ID: P10627) and (2) the interaction between Twist and Zelda.

### 4.2.2 APPLICATION TO MORTALITY RISK DATA

We next applied DeepROCK to the relationship between mortality risk factors and long-term health outcomes in the US population. We used a mortality dataset from the National Health and Nutrition Examination Survey (NHANES I) and NHANES I Epidemiologic Follow-up Study (NHEFS) (Cox et al., 1997). The dataset examined $n = 14,407$ participants in the US between 1971 and 1974 by $p = 79$ clinical and laboratory measurements. The dataset also reported the mortality status of

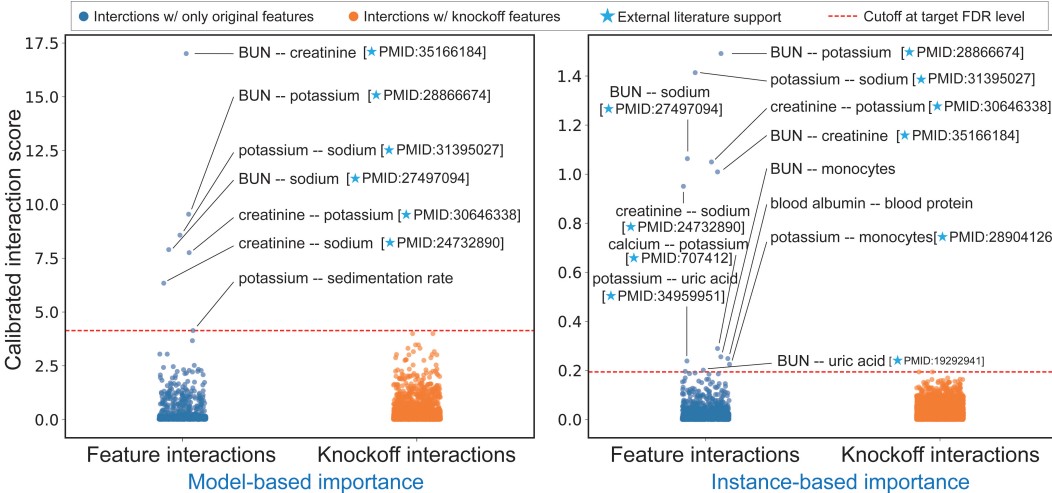

Figure 4: Applying DeepROCK to the mortality risk data using both model-based and instance-based importance measures. Though DeepROCK only induces a ranking on feature interactions, knockoff-involving interactions are also provided for comparison. The feature interactions identified by DeepROCK at the target FDR level are labeled with the corresponding mortality risk factors. The blue stars indicate the interactions supported by literature evidence in terms of their PubMed identifiers.

participants as of 1992 to trace whether they had died or were still alive, where $4,785$ individuals had died before 1992.

We evaluated the identified mortality risk factor interactions by DeepROCK at the target FDR level using both model-based and instance-based importance measures. In the two different settings, 6 out of 7 and 10 out of 12 interactions are supported by literature evidence, with PubMed identifiers are shown in Figure 4. For example, it is known that the blood urea nitrogen (BUN)/creatinine ratio is nonlinearly associated with all-cause mortality and linearly associated with cancer mortality (Shen et al., 2022). Additionally, the BUN/potassium interaction can be justified by combining the following two facts: (1) BUN level is associated with the chronic kidney disease development (Collins et al., 2017), and (2) mortality rate progressively increases with abnormal potassium levels in patients with chronic kidney diseases (Seki et al., 2019).

## 5  DISCUSSION AND CONCLUSION

In this work, we have proposed a novel method, DeepROCK, that can help to interpret a deep neural network model by detecting relevant, non-additive feature interactions, subject to FDR control. FDR control is achieved by using knockoffs that perfectly mimic the empirical dependence structure among the original features. Together with the knockoffs, DeepROCK employs a novel DNN architecture, namely, a plugin pairwise-coupling layer, to maximize statistical power by encouraging the competition of each feature against its knockoff counterpart during training. Through simulation studies, we discovered surprisingly that using existing importance measures does not correctly control the FDR. To resolve this issue, DeepROCK proposes a calibration procedure applied to existing interaction importance measures to control the FDR at a target level. Our experiments demonstrate that DeepROCK achieves FDR control with high power on both simulated and real datasets.

This work points to several promising directions for future research. First, DeepROCK is designed for feedforward DNNs. Extending our method to other DNN models such as CNNs and RNNs would be interesting directions to pursue. Second, we observe that instance-based importance consistently achieves much higher power with more conservative FDR estimation than model-based importance. We would like to better understanding the reason for this trend. Finally, DeepROCK is limited to pairwise interaction detection. Supporting higher-order interaction detection with controlled FDR is critical in explaining genetic mechanisms, diseases, and drug effects in healthcare domains.

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

# A APPENDIX

| | |
|---|---|
| $F_1$ | $\pi^{x_1 x_2}\sqrt{2x_3} - \sin^{-1}(x_4) + \log(x_3 + x_5) - \frac{x_9}{x_{10}}\sqrt{\frac{x_7}{x_8}} - x_2 x_7$ |
| $F_2$ | $\pi^{x_1 x_2}\sqrt{2|x_3|} - \sin^{-1}(0.5x_4) + \log(|x_3 + x_5| + 1) - \frac{x_9}{1+|x_{10}|}\sqrt{\frac{x_7}{1+|x_8|}} - x_2 x_7$ |
| $F_3$ | $\exp|x_1 - x_2| + |x_2 x_3| - x_3^{2|x_4|} + \log(x_4^2 + x_5^2 + x_7^2 + x_8^2) + x_9 + \frac{1}{1+x_{10}^2}$ |
| $F_4$ | $\exp|x_1 - x_2| + |x_2 x_3| - x_3^{2|x_4|} + (x_1 x_4)^2 + \log(x_4^2 + x_5^2 + x_7^2 + x_8^2) + x_9 + \frac{1}{1+x_{10}^2}$ |
| $F_5$ | $\frac{1}{1+x_1^2+x_2^2+x_3^2} + \sqrt{\exp(x_4 + x_5)} + |x_6 + x_7| + x_8 x_9 x_{10}$ |
| $F_6$ | $\exp(|x_1 x_2| + 1) - \exp(|x_3 + x_4| + 1) + \cos(x_5 + x_6 - x_8) + \sqrt{x_8^2 + x_9^2 + x_{10}^2}$ |
| $F_7$ | $(\arctan(x_1) + \arctan(x_2))^2 + \max(x_3 x_4 + x_6, 0) - \frac{1}{1+(x_4 x_5 x_6 x_7 x_8)^2} + (\frac{|x_7|}{1+|x_9|})^5 + \sum_{i=1}^{10} x_i$ |
| $F_8$ | $x_1 x_2 + 2^{x_3+x_5+x_6} + 2^{x_3+x_4+x_5+x_7} + \sin(x_7 \sin(x_8 + x_9)) + \arccos(0.9x_{10})$ |
| $F_9$ | $\tanh(x_1 x_2 + x_3 x_4)\sqrt{|x_5|} + \exp(x_5 + x_6) + \log((x_6 x_7 x_8)^2 + 1) + x_9 x_{10} + \frac{1}{1+|x_{10}|}$ |
| $F_{10}$ | $\sinh(x_1 + x_2) + \arccos(\tanh(x_3 + x_5 + x_7)) + \cos(x_4 + x_5) + \sec(x_7 x_9)$ |

Table A.1: A test suite of data-generating simulation functions by Tsang et al. (2018a).

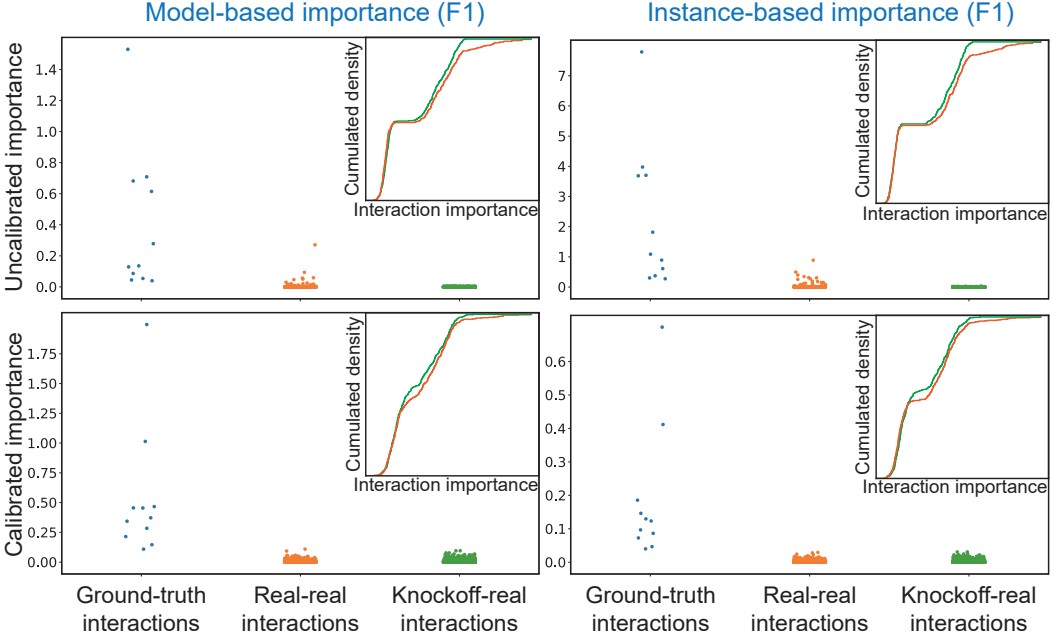

Figure A.1: Interaction score comparison between the real-real interactions and knockoff-real interactions.

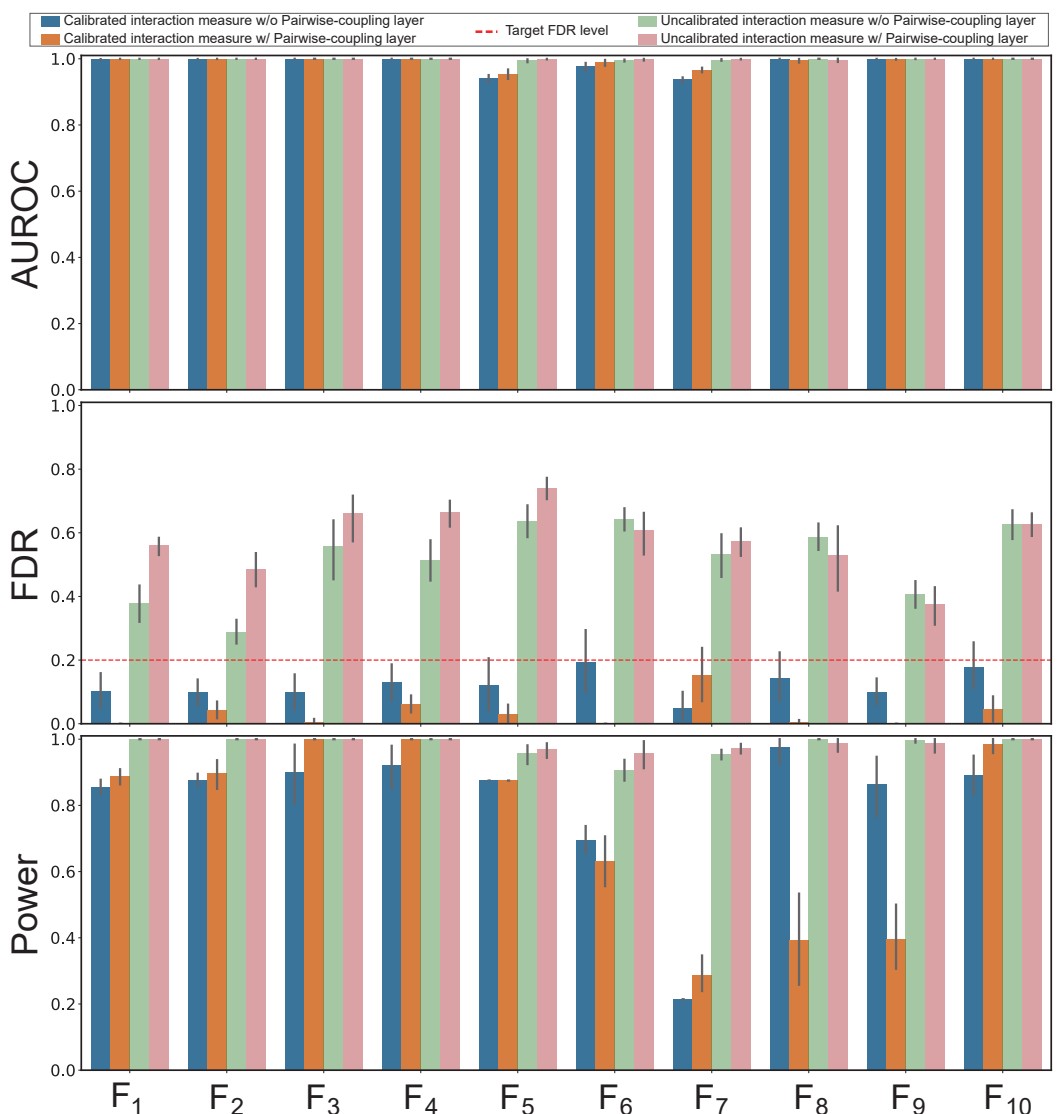

Figure A.2: Evaluating DeepROCK on simulated datasets using SmoothGrad-based importance measures.

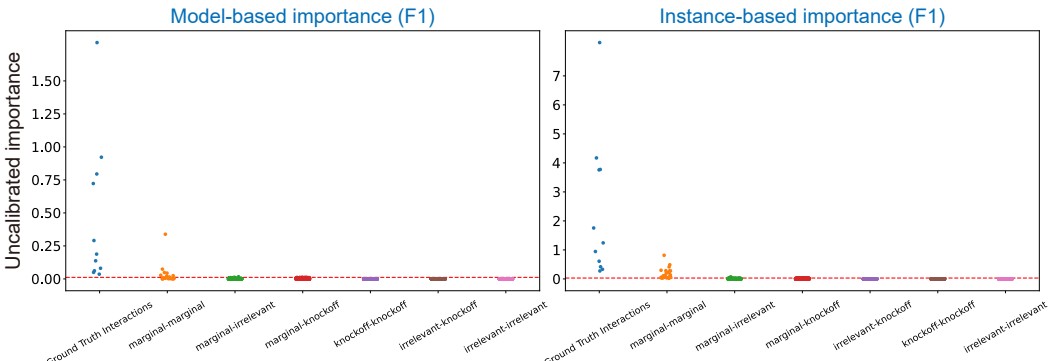

Figure A.3: Interaction score distribution organized in a finer categorization.

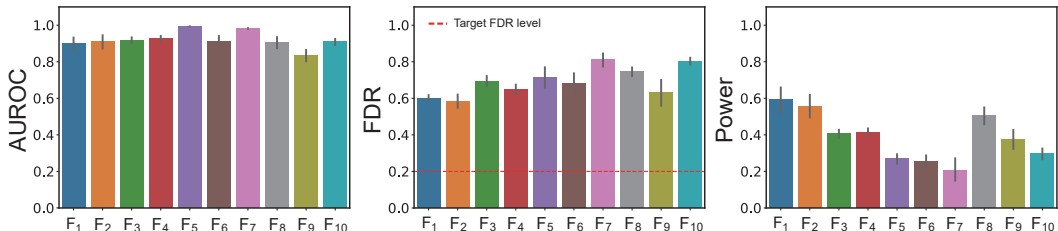

Figure A.4: Evaluating the baseline method (permutation-based p-value Cui et al. (2022) coupled with the Benjamini-Hochberg procedure Benjamini & Hochberg (1995)) on simulated datasets.

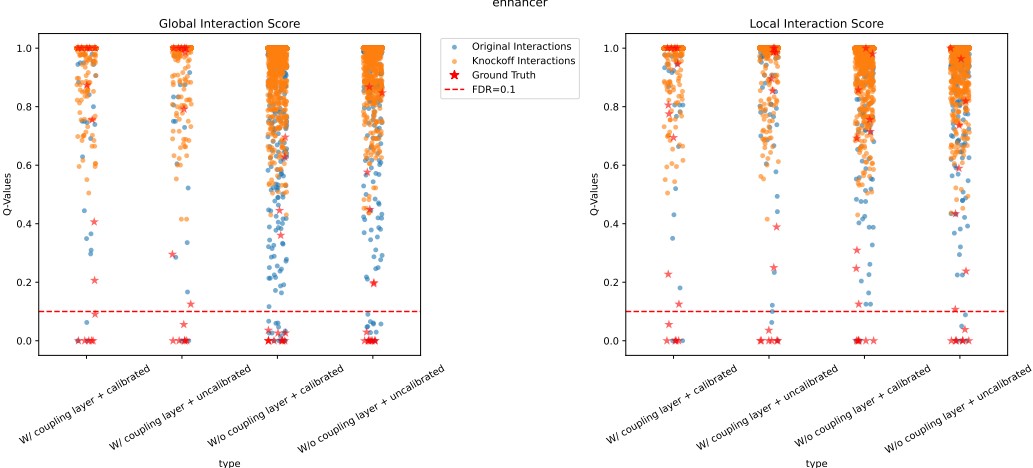

Figure A.5: Applying DeepROCK to the Drosophila enhancer data in the absence of calibration or a coupling layer.

