# OpenReview forum: "DeepROCK: Error-controlled interaction detection in deep neural networks"
_ICLR.cc/2024/Conference — Submitted to ICLR 2024_

### Official Review · Reviewer_JPER · 2023-10-29

**Soundness:** 3 good
**Presentation:** 3 good
**Contribution:** 3 good
**Rating:** 6
**Confidence:** 3

**Summary:**

- The authors measure the false discovery rate (FDR) of existing interaction detection methods in DNNs to quantify their error rate.
- They use knockoff features to overcome the lack of p-values.
- The main contribution is the combination of knockoff framework and interaction detection algorithms. Specifically, they introduce DeepROCK, which entails a novel architecture including a pairwise-coupling layer and a calibration procedure, allowing to control the error rate.
- The authors run experiments in simulated and real-world scenarios to demonstrate the effectiveness of DeepROCK.

**Strengths:**

- The authors address a very relevant topic, namely the detection of feature interaction in DNNs, along with a procedure to control the error rate.
- They propose an interesting idea to approach the problem, which is the connection of knockoff framework and interaction detection algorithms to control FDR. Ultimately, this makes interaction detection algorithms useful in high-stake applications.
- Sound presentation of their approach and required mathematical background knowledge.
- Meaningful experiments both with simulated data and two real-world datasets.

**Weaknesses:**

- For the real-world experiments in Fig. 3 and 4, there is no comparison with existing methods. It would be interesting to study found interactions without calibration/coupling layer.
- (nitpick) typos in section 2.2: “withcovariance”

**Questions:**

- Will the code be published for reproducibility?

---

> ### Author Response · Authors · 2023-11-21
> **Response to reviewer JPER**
>
> We thank the reviewer for their constructive comments. Please allow us to make some clarifications below.
>
> 1. The reviewer suggested a comparison of DeepROCK with existing methods. As the only baseline method available to us, we employed the Benjamini–Hochberg procedure[1] to achieve false discovery rate (FDR) control. This is done by working with p-values calculated using a permutation procedure tailored for neural networks to assess the significance of interactions[2]. As shown in Figure A.4 in the paper appendix, the permutation-based p-value, coupled with the Benjamini–Hochberg procedure, fails to correctly control the false discovery rate (FDR), rendering the only available baseline method invalid. It is important to note that currently, there is no valid existing method specifically designed for error-controlled explanations in deep neural networks. Addressing this specific problem makes DeepROCK's contribution noteworthy.
>
> 2. The reviewer inquired about the detected interactions in the absence of calibration or a coupling layer. In response to this question, we ran an ablation study in which we modified DeepROCK to exclude either the calibration or the coupling layer. To compare the four different settings, we quantified each interaction in each setting using the averaged q-value across 20 runs. The q-value is defined as the minimum false discovery rate at which an observed score is deemed significant. We applied the modified DeepROCK to the _Drosophila_ enhancer data, for which we have a list of well-characterized interactions in early _Drosophila_ embryos, as summarized by [3], serving as the ground truth. Despite acknowledging that some of the settings fail to control false discovery rate (FDR) or underperform in detection power, we found that all four settings demonstrated proficiency in prioritizing the ground-truth interactions, as illustrated in Figure A.5 in the paper appendix. Furthermore, we observed that the removal of the coupling layer resulted in a reduction in the separation between the ground-truth interactions and others, as expected.
>
> 3. The reviewer inquired about the release of the code for reproducibility. Due to the anonymous submission requirement, we have not included the GitHub repository link. However, we assure you that the code and data needed to reproduce the experimental results are prepared and will be made publicly available upon the acceptance of the paper.
>
> __References__
>
> [1] Y. Benjamini and Y. Hochberg. Controlling the false discovery rate: a practical and powerful approach to multiple testing. Journal of the Royal Statistical Society Series B, 57:289–300, 1995.
>
> [2] T. Cui, K. El Mekkaoui, J. Reinvall, A. S. Havulinna, P. Marttinen, and S. Kaski. Gene–gene interaction detection with deep learning. Communications Biology, 5(1):1238, 2022.
>
> [3] S. Basu, K. Kumbier, J. B. Brown, and B. Yu. Iterative random forests to discover predictive and stable high-order interactions. Proceedings of the National Academy of Sciences, 115(8):1943–1948, 2018.

---

### Official Review · Reviewer_u7Lb · 2023-11-01

**Soundness:** 2 fair
**Presentation:** 2 fair
**Contribution:** 3 good
**Rating:** 3
**Confidence:** 2

**Summary:**

The paper uses knockoffs to control false discovery rate better in discovering interactions. Given existing ways to measure how much a model depends on the interaction between two feature, the key steps are to produce a calibrated score and feature-interaction rank threshold to improve FDR control of interactions while not losing power. The paper has useful experiments.

**Strengths:**

1. The problem of detecting interactions is important for science.
2. The paper's experiments show clear advantage over existing methods in terms of power and FDR.
3. The need for calibrated interaction scores is surprising.

**Weaknesses:**

As the main goal is variable selection and the stated goal is FDR control, it seems necessary that there should be a proof of FDR control. To start here, one example of a definition of an important feature is $Y \perp X_j \mid X_{-j}$. Is there a version of this  in terms of interaction ? Possibly, the following $$Y \perp (X_j, X_i) \mid X_{-ji}, (E[Y \mid X_j], E[Y\mid X_i]) $$

Without connecting such a definition to the how you are using the knockoffs framework, I cannot trust a claim about FDR control.  I see a few things that could help, if the knockoff swap property holds, then real-real interactions and knockoff-real interactions also should satisfy the swap property. I think this should be shown but it seems believable.

But then it should be made clear that the flip property is satisfied for the interaction measures in some sense. Otherwise, the knockoff based selection would not provide FDR control.

Happy to discuss further and increase score.

**Questions:**

See earlier sections.

Beyond those,

1. It has been suggested that integrated gradients do not have fidelity when it comes to explaining models. Then, what kind of conclusions can I make from scores based on them ?

2.  The model-dependent score seems to be archictecture specific. Are there concerns about multiplying weight matrices across layers in, for example, deep residual networks?

3. Is there something formal to understand this better "Intuitively, the interaction between two marginally important features naturally has a higher importance score than random interactions, even though they are all false"?

---

> ### Author Response · Authors · 2023-11-21
> **Response to reviewer u7Lb (part 1/2)**
>
> We thank reviewer for their constructive comments. Please allow us to make some clarifications below.
>
> 1. The reviewer suggested defining an important feature, denoted as $j$, through the condition $Y \perp X_j| X_{-j}$ and inquired about the existence of an analogous definition for important interactions. First, the proposed definition appears to be problematic because, intuitively, an important feature should not be conditionally independent of the response variable $Y$. We reasoned that the reviewer's definition should be modified and applied to a set of important features, denoted as $J \subset \left \\{ 1, 2, \ldots, p \right \\}$, such that, conditioned on $J$, the complementary set of features, $J^c = \left \\{ 1, 2, \ldots, p \right \\}\backslash J$, is independent of the response variable $Y$, i.e., $Y \perp X_{J^c}| X_{J}$. We value the mathematical rigor exhibited by the reviewer and highlight the corresponding definition for significant interactions outlined in Section 2.1, under the heading "Problem Setup". Specifically, we denote the set of important interactions as $\mathcal{S}\subset J \times J$, such that, conditioned on $\mathcal{S}$ and $J$, the response $Y$ is independent of interactions in the complement $\mathcal{S}^c=\\{1,\cdots, p\\}\times \\{1,\cdots, p\\}\backslash \mathcal{S}$, i.e., $Y \perp X_{\mathcal{S}^c}| X_{J}, X_{\mathcal{S}}$. It is worth noting that we regard the features engaged in significant interactions as marginally important; however, a marginally important feature may not necessarily participate in a significant interaction.
> Here is an illustrative example supporting the claim: $Y = f(X_1) + g(X_2, X_3)$.
>
> 2. The reviewer asked about the existence of a swap property for real-real interactions and knockoff-real interactions, analogous to the knockoff swap property. We do not believe that the swap property directly applies to the interaction setting. If such a swap property exists, then using off-the-shelf feature interaction importance measures should, in theory, adequately control the false discovery rate (FDR). However, as demonstrated in our paper, this is not the case unless we implement the calibration procedure (Section 4.1 under the heading "Performance on simulated data"). Our claim is further substantiated by the disparity in score distribution between the non-ground-truth real-real interactions and knockoff-real interactions. As shown in Figure A.1 in the paper appendix, the uncalibrated non-ground-truth real-real interactions and knockoff-real interactions exhibit a notable distributional disparity, suggesting a lack of adherence to the swap property. Moreover, Figure A.1 indicates that the calibration effectively mitigates this distribution disparity, thereby enhancing the practical utility of knockoff-real interactions as a control for estimating the false discovery rate (FDR). It is also noteworthy that, based on the cumulative distribution function, the calibration has not completely eliminated the distribution disparity between real-real interactions and knockoff-real interactions. Investigating alternative calibration strategies would be an intriguing avenue for further exploration.

---

> ### Author Response · Authors · 2023-11-21
> **Response to reviewer u7Lb (part 2/2)**
>
> 3. The reviewer raised concerns about the reliability of using integrated gradient-based interaction importance for model explanations and subsequently questioned the validity of the conclusions we drew. First, it is essential to note that we employ expected gradients[1], a state-of-the-art method refined from integrated gradients.  Importantly, expected gradients is guaranteed to adhere to the interpretability axioms. Second, we concur with the reviewer that relying on a single method is not advisable, because no method is flawless. We therefore propose here an alternative, instance-based importance measure by substituting expected gradients with SmoothGrad[2] another state-of-the-art and commonly used method. As shown in Figure A.2 in the paper appendix, the interaction importance derived from SmoothGrad produces results similar to those obtained from the expected gradient-based method. These findings, coupled with the results from model-based interaction importance, not only reinforce the conclusions drawn from the expected gradient-based method but also underscore the robustness of DeepROCK.
>
> 4. The reviewer raised a question about whether the model-based importance is applicable only to MLPs or if it extends to non-MLP architectures such as deep residual networks. While we acknowledge that the model-based importance is specific to MLPs, we want to emphasize that our intention was not to address all architectures using this method. Instead, we introduced the model-based importance as an example to showcase the robustness of DeepROCK across various types of importance scores, similar to the previously mentioned SmoothGrad-based interaction importance. From a practitioner's perspective, we recommend instance-based importance measures because they are model-agnostic.
>
> 5. The reviewer inquired about the formal understanding of the assertion that the interaction between two marginally important features naturally yields a higher importance score than random interactions even though they have no interaction. First, we began with the uncalibrated interaction scores presented in Figure A.1 in the paper appendix and organized the interactions into distinct groups. To be specific, utilizing the ground truth labels, we partitioned the original features into two groups (marginal and irrelevant) based on their marginally important status, resulting in a total of seven interaction groups, as depicted in Figure A.3 in the paper appendix. From the figure, we observe that the uncalibrated and non-ground-truth interactions between two marginally important features are substantially higher than other non-ground-truth interactions, which could potentially pose challenges for FDR estimation. Second, we offer an explanation for the biased importance of interactions between two marginally important features in the setting of model-based importance. As depicted in Section 3.2 of the paper, under the heading "Feature interaction importance", the model-based interaction importance is determined by two factors:
> (1) the relative importance between the original feature and its knockoff counterpart, and
> (2) the relative importance among all features.
> These two factors are inherently large for marginally important features by definition, even if they do not participate in any interactions.
>
>
> __References__
>
> [1] G. Erion, J. D. Janizek, P. Sturmfels, S. M. Lundberg, and S.-I. Lee. Improving performance of deep learning models with axiomatic attribution priors and expected gradients. Nature Machine Intelligence, 3(7):620–631, 2021.
>
> [2] D. Smilkov, N. Thorat, B. Kim, F. Viégas, and M. Wattenberg. Smoothgrad: removing noise by adding noise. arXiv preprint arXiv:1706.03825, 2017.

---

> > ### Author Response · Authors · 2023-11-27
> > **Response to reviewer u7Lb**
> >
> > We want to thank reviewer again for thoughtful review and constructive feedback.
> >
> > We have carefully considered and addressed each of the concerns and suggestions the reviewer raised. Our team believes that these changes have significantly improved the overall quality of our work. We are thankful for the opportunity to engage in this dialogue with the reviewer, as it has undoubtedly strengthened our submission. However, we noticed that the review status has not been updated since our revision. We understand that the reviewer has a busy schedule, and we truly appreciate the time and effort the reviewer has invested in the reviewing process. If the reviewer has had a chance to reevaluate our work, we would be keen to hear more thoughts and any further suggestions from the the reviewer. If there are any additional details or information the reviewer might require, please don't hesitate to let us know. Thanks once again for your dedication to the review process.

---

### Official Review · Reviewer_f17i · 2023-11-24

**Soundness:** 3 good
**Presentation:** 3 good
**Contribution:** 2 fair
**Rating:** 6
**Confidence:** 3

**Summary:**

[Note on review timing: unfortunately I was only assigned this paper yesterday, after the reviewer-author discussion period closed. However I have made sure to read the authors' responses to the other reviews.]
The paper introduces DeepROCK, a method for detecting 'feature interactions' when interpreting a neural network. The knockoffs framework and a novel architecture is used to control the false discovery ratio (FDR). Empirical results show that the method is able to identify pairwise interactions in toy and real-world datasets.

**Strengths:**

+ Interesting integration of knockoffs for FDR control in DNNs.
+ Addresses a critical need for interpretable and reliable DNN predictions.
+ Provides empirical evidence demonstrating the potential of the approach.

**Weaknesses:**

+ The generality of the method across different DNN architectures in not developed. In fact, the method only seems to work with MLPs.
+ The method seems somewhat heuristic. As pointed out by reviewer 1, the sentence 'Intuitively, the interaction between two marginally important features naturally has a higher importance score than random interactions' is used to motivate the calibration in section 3.2, but is not very well formalized.
+ The method seems very specialized to pairwise interactions, and it's not obvious if the method would scale to $n$-wise interactions without a significant cost in computational complexity.
+ As I understand it, there are no statistical guarantees due to the use of function approximation in the KnockoffGAN and MLP.

**Questions:**

See weaknesses

---

> ### Author Response · Authors · 2023-11-25
> **Response to reviewer f17i (part 1/2)**
>
> We thank reviewer for their constructive comments. Please allow us to make some clarifications below.
> 1. The reviewer expressed concerns regarding the generalizability of DeepROCK to DNN architectures other than MLP. It is crucial to clarify that DeepROCK is designed to be applicable to any off-the-shelf DNN architecture beyond MLP. We recommend using DeepROCK with instance-based importance measures because they are model-agnostic. While we acknowledge that model-based importance is specific to MLPs, it’s important to note that our intention was not to address all architectures using this particular method. Instead, we introduced model-based importance as an illustrative example to demonstrate the robustness of DeepROCK across various types of importance scores.
>
> 2. The reviewer raised concerns about the motivation behind feature interaction importance calibration and sought a formal understanding of the claim that the interaction between two marginally important features inherently results in a higher importance score than random interactions, even in the absence of any genuine interaction. First, we began with the uncalibrated interaction scores in Simulation Function 1 and organized the interactions into distinct groups. To be specific, utilizing the ground truth labels, we partitioned the original features into two groups (marginal and irrelevant) based on their marginally important status, resulting in a total of seven interaction groups, as depicted in Figure A.1 in the paper appendix. From the figure, we observe that the uncalibrated and non-ground-truth interactions between two marginally important features are substantially higher than other non-ground-truth interactions, which could potentially pose challenges for FDR estimation. Second, we offer an explanation for the biased importance of interactions between two marginally important features in the setting of model-based importance. As depicted in Section 3.2 of the paper, under the heading “Feature interaction importance”, the model-based interaction importance is determined by two factors: (1) the relative importance between the original feature and its knockoff counterpart, and (2) the relative importance among all features. These two factors are inherently large for marginally important features by definition, even if they do not participate in any interactions.
>
> 3. The reviewer raised concerns about the practicality of FDR-controlled pairwise interactions, highlighting uncertainty in their scalability to higher-order interactions. First, the detection of pairwise interactions holds practical significance in various biological contexts, including gene-gene, gene-disease, gene-drug, and gene-environment interactions. These interactions are crucial for understanding genetic mechanisms, diseases, and drug effects. Furthermore, it’s important to note that, currently, there is no valid existing method specifically designed for error-controlled explanations in deep neural networks. Therefore, addressing this specific gap in the literature makes DeepROCK’s contribution particularly noteworthy. Finally, similar to how FDR control in pairwise interaction detection necessitates first-order feature importance for calibration, we posit that DeepROCK’s calibration of pairwise interaction detection paves the way for extending FDR control to higher-order interactions. Exploring the extension of DeepROCK to include FDR-controlled higher-order interaction detection would be an intriguing avenue for further investigation.

---

> ### Author Response · Authors · 2023-11-25
> **Response to reviewer f17i (part 2/2)**
>
> 4. The reviewer expressed concerns about the absence of statistical guarantees arising from the use of function approximation in both KnockoffGAN and MLP. First, it is crucial to clarify that both KnockoffGAN [1][2][3] and MLP [4] have established a track record of controlling the FDR with statistical guarantees, even in the presence of function approximation. Moreover, the selection of KnockoffGAN and MLP is further substantiated by the comprehensive simulation studies presented in the paper (Section 4.1, under the heading “Performance on simulated data”). Finally, it’s crucial to acknowledge that there is currently no established method specifically designed for error-controlled explanations in deep neural networks. Therefore, any effective methods in this realm would be highly significant. We believe that DeepROCK represents just the beginning of a more extensive effort to further develop and enhance this concept with a better theoretical understanding.
>
> __References__
>
> [1] J. Jordon, J. Yoon, and M. van der Schaar. KnockoffGAN: Generating knockoffs for feature selection using generative adversarial networks. In International Conference on Learning Representations, 2018.
>
> [2] Y. Romano, M. Sesia, and E. Candès. Deep knockoffs. Journal of the American Statistical Association, 115(532):1861–1872, 2020.
>
> [3] M. Sudarshan, W. Tansey, and R. Ranganath. Deep direct likelihood knockoffs. In Advances in Neural Information Processing Systems, volume 33, pages 5036–5046, 2020.
>
> [4] Y. Y. Lu, Y. Fan, J. Lv, and W. S. Noble. DeepPINK: reproducible feature selection in deep neural networks. In Advances in Neural Information Processing Systems, 2018.

---

> > ### Comment · Reviewer_f17i · 2023-11-26
> >
> > Thank you for your prompt and comphrehensive reply. Let me make sure I understand a few points in your response:
> >
> > 1. You say 'DeepROCK is designed to be applicable to any off-the-shelf DNN architecture beyond MLP', and that the model-based importance scores are the main MLP-specific elements. That makes sense. Looking back, I misunderstood figure 1, which states 'Overview of DeepROCK. DeepROCK is built upon an MLP...'. However, of course the pairwise-coupling layer could also be applied on top of e.g. a CNN. I think it would make sense to reword this to emphasise that the choice of MLP is not integral to the DeepROCK method.
> > 2. Thanks for clarifying the empirical argument.
> > 3. I understand that detecting pairwise interactions is useful in discerning patterns in biological data analysis. My concern is about the approach of training an unconstrained neural network and then trying to infer pairwise interactions. Specifically, you say in one of the experimental sections 'Intuitively, if there is a strong interaction between TF1 and TF2, and between TF2 and TF3, then a high interaction score will also be expected between TF1 and TF3, even if there is no direct interaction between them'. Given this, I wonder how it's possible to discern which of the interactions with high score are 'true' interactions and which are spurious due to this phenomenon.
> > 4. Thanks for elaborating on the significance of the approach and the discussion about statistical guarantees.
> >
> > Based on the authors' responses, I will raise my score.

---

> > > ### Author Response · Authors · 2023-11-26
> > > **Response to Reviewer f17i**
> > >
> > > We thank reviewer for your constructive suggestions. Please allow us to make some clarifications below.
> > > 1. The reviewer recommended a rephrasing of the description of DeepROCK to mitigate potential confusion. We acknowledge and appreciate the reviewer’s suggestion, recognizing the importance of enhancing the clarity regarding the generalizability of DeepROCK. In the upcoming revised version, we are committed to implementing the recommended changes, specifically in the redesign of Figure 1 and the rephrasing of the DeepROCK description. This revision will explicitly highlight that DeepROCK is designed to be applicable to any off-the-shelf DNN architecture beyond MLP.
> > > 2. The reviewer expressed curiosity about potential methods for eliminating transitive interactions from the detected interactions. First, we reasoned that under the current setup of DeepROCK, which is designed to detect non-additive feature interactions from the data, it is unlikely to eliminate transitive interactions. Here is an illustrative example supporting the claim: $Y=f(X1,X2)+g(X2,X3)$, where the feature interaction $(X1,X3)$ represents both a transitive and non-additive interaction. In other words, under the current setup of DeepROCK, $(X1,X3)$ is classified as a valid non-additive feature interaction, even though it may not align with our intended criteria. Second, we consider the reviewer’s suggestion regarding the elimination of transitive interactions highly valuable, particularly in the context of uncovering patterns in biological data analysis. We concur with the reviewer that additional information is necessary to differentiate transitive interactions from genuine interactions. If we had FDR-controlled higher-order interaction detection, we could potentially deduce that there is no tertiary interaction $(X1,X2,X3)$ in the data. This conclusion could, in turn, offer insights into the potential existence of false pairwise interactions.

---

### Meta-Review · Area_Chair_oqD2 · 2023-12-13

**Metareview:**

The paper presents a knockoff based approach for finding interactions in neural networks. The problem is interesting as noted by the two reviewers that are weakly positive and important. The solution taken building off knockoffs has potential as well. However for a paper concerned with the false discovery rate, which is defined with respect to a null hypothesis, there is little in the paper that discusses what the null hypothesis is for the interactions. This was the main concern/confusion of the negative reviewer, which I agree with from my quick read as well. The reviewer and the authors had a discussion about this issues, but the definition raised in the rebuttal has flaws noted by the reviewer in a further reply. Overall, I'm positive on the paper, but I strongly encourage the reviewers to tackle this issues/confusion.

**Justification For Why Not Higher Score:**

Papers based for hypothesis testing need to have a very clear statement of the null hypothesis and a corresponding discussion of FDR/type-1 error control.

**Justification For Why Not Lower Score:**

N/A

---

### Decision · Program_Chairs · 2024-01-16

Reject